# Learning Human Action Recognition Representations Without Real Humans

**Howard Zhong[1,2], Samarth Mishra[2,3], Donghyun Kim[2,6], SouYoung Jin[4], Rameswar Panda[2], Hilde Kuehne[2,5], Leonid Karlinsky[2], Venkatesh Saligrama[3], Aude Oliva[1,2], Rogerio Feris[2]**

[1]MIT, [2]MIT-IBM Watson AI Lab, [3]Boston University, [4]Dartmouth College, [5]Goethe University, [6]Korea University

## Abstract

Pre-training on massive video datasets has become essential to achieve high action recognition performance on smaller downstream datasets. However, most large-scale video datasets contain images of people and hence are accompanied with issues related to privacy, ethics, and data protection, often preventing them from being publicly shared for reproducible research. Existing work has attempted to alleviate these problems by blurring faces, downsampling videos, or training on synthetic data. On the other hand, analysis on the *transferability* of privacy-preserving pre-trained models to downstream tasks has been limited. In this work, we study this problem by first asking the question: can we pre-train models for human action recognition with data that does not include real humans? To this end, we present, for the first time, a benchmark that leverages real-world videos with *humans removed* and synthetic data containing virtual humans to pre-train a model. We then evaluate the transferability of the representation learned on this data to a diverse set of downstream action recognition benchmarks. Furthermore, we propose a novel pre-training strategy, called Privacy-Preserving MAE-Align, to effectively combine synthetic data and human-removed real data. Our approach outperforms previous baselines by up to 5% and closes the performance gap between human and no-human action recognition representations on downstream tasks, for both linear probing and fine-tuning. Our benchmark, code, and models are available at https://github.com/howardzh01/PPMA.

## 1 Introduction

Action recognition is the task of classifying human actions from video sequences [9, 8, 42, 54, 70]. Pre-training action recognition models on large-scale datasets significantly improve the accuracy of these models, especially when generalizing to downstream tasks with limited data [1, 29, 15]. Furthermore, with the advent of transformers in vision [25, 69], pre-training on large-scale datasets has become increasingly important for high performance [36].

While the use of large-scale datasets can yield significant accuracy improvements, it poses several problems. First, it is costly and time-consuming to curate and annotate these large-scale real-world datasets. Unsupervised [50] and semi-supervised pre-training [2] can leverage large datasets without the cost of labeling, but it does not address the privacy and ethical concerns related to pre-training with large datasets. Visual characteristics such as skin tone and gender can lead to cognitive biases in models trained from large-scale datasets. It is difficult to control for these biases [10, 74]. Furthermore, many large-scale datasets contain videos including humans, who may be identifiable based on their face, clothing, or other sensitive attributes. In particular, most existing video datasets contain images of people that were collected *without consent*. This is a legal issue, as personal data is protected by legislation such as GDPR [3]. Moreover, it is difficult to control undesirable biases related to gender

37th Conference on Neural Information Processing Systems (NeurIPS 2023) Track on Datasets and Benchmarks.

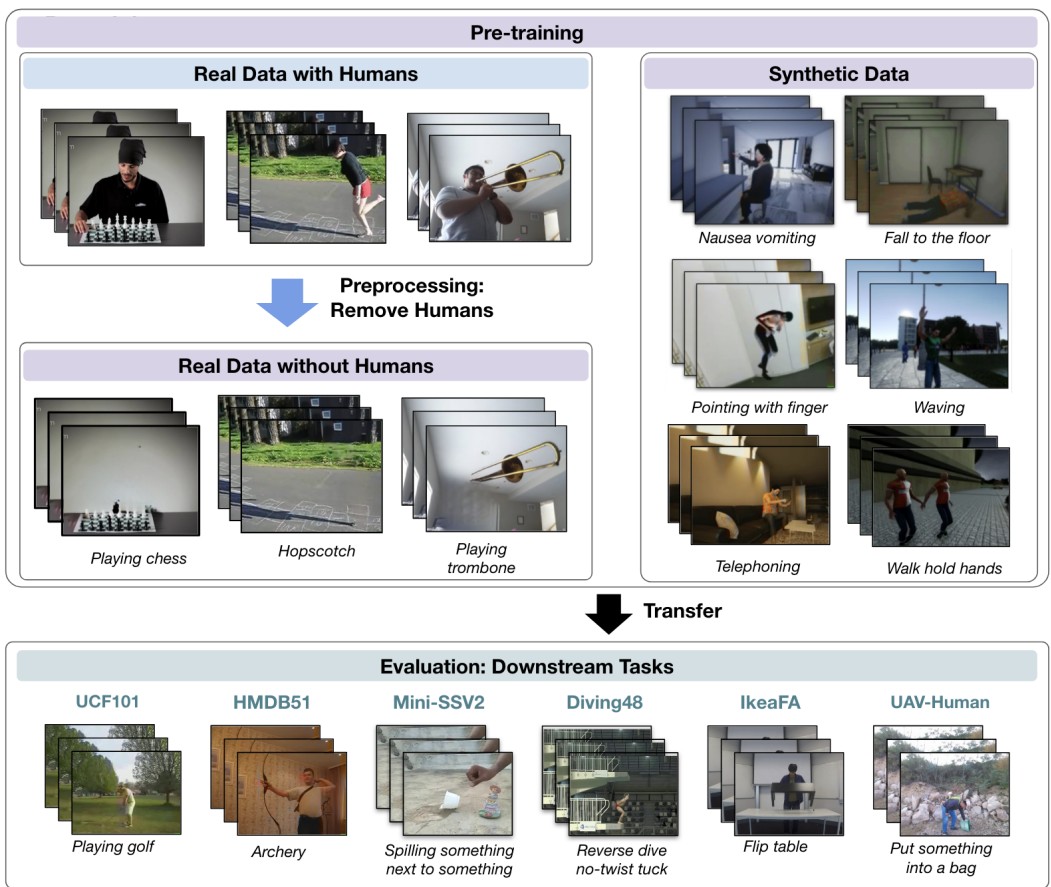

Figure 1: **Our privacy-preserving action recognition benchmark.** Real-world videos are initially pre-processed to remove humans and then combined with synthetic data containing virtual humans for pre-training. The transferability of these pre-trained models are then evaluated on a diverse set of downstream human action recognition tasks.

and race in existing large-scale datasets. As an example, state-of-the art models may fail to predict actions such as "woman snowboarding" [39], given that the training set contains more videos of men performing this action. Finally, membership reconstruction attacks retrieve training data from a trained model and can expose identifying information [35, 62].

Existing works have proposed various ways to alleviate these problems. Methods include downsampling [21], obfuscation [11, 82], and adversarial training to modify the image [22, 58, 77]. However, these works do not analyze the transferability of the representations in pre-trained models to downstream tasks. As membership reconstruction attacks that recover training data from a pre-trained model are becoming increasingly sophisticated [35, 62], there is a growing need to determine how to leverage privacy-preserving data to pre-train models. The PASS dataset [5] was recently created to study representations learned from images without personally identifiable information for image classification.

The goal of this work is to pre-train representations for human action recognition without *real humans* in the data. SynAPT [75] is one of the few works that address this privacy-preserving transferability problem but only explores the use of synthetic data. It proposed a benchmark to evaluate the transferability of action recognition features pre-trained on synthetic data on different downstream tasks. SynAPT showed that pre-training on synthetic data was useful for downstream tasks with low scene-object bias but not as effective for tasks with high scene-object bias (i.e., when action categories are likely to be recognized solely by the scene or objects in the background, instead of the temporal dynamics of the action itself). High scene-object bias tasks require better real-world embeddings of the context surrounding the action, but this may not necessarily require the human to

be present in the video. In this work, we demonstrate that we can solve this problem while preserving privacy by obfuscating humans in real videos.

We pre-train models with a *combination of synthetic videos and human-removed real videos* so that the synthetic data will teach the model temporal dynamics of the action while the human-removed real data will help it learn contextual features in the scene. To the best of our knowledge, no previous work has analyzed the transferability of action recognition features trained on a mix of synthetic data and real data without humans on downstream tasks.

As shown in Fig. 1, we extend the SynAPT benchmark to address this important problem. We use a pre-training dataset consisting of the synthetic videos from SynAPT and human-removed real videos from No-Human Kinetics. This consists of videos in the Kinetics dataset [42] where the humans are removed from each frame by applying the HAT framework [19] to first detect the human segmentation mask and then inpaint within this mask. We evaluate the model on the same six diverse, downstream tasks from the SynAPT benchmark.

We then analyze how to best leverage synthetic data and human-removed real data to pre-train the models for different downstream tasks. Vision transformers [4, 8] yield state-of-the-art performance on human action recognition benchmarks but require extensive pre-training on large datasets such as ImageNet-21k [61]. As datasets of such scale usually contain humans, we require a more data-efficient method to pre-train models on the smaller, privacy-preserving datasets. Based on the Masked Autoencoders (MAE) method [37, 67, 30], we propose a two-stage pre-training strategy we denote as *Privacy-Preserving MAE-Align (PPMA)*. We first perform the data-efficient self-supervised MAE pre-training on No-Human Kinetics to learn contextual features. We then conduct supervised pre-training on both No-Human Kinetics and synthetic data through action classification to align the representations learned from MAE to action labels. We found both stages of pre-training were crucial to achieve high performance and able to significantly outperform the methods in SynAPT [75] across the six downstream tasks.

Furthermore, when applying our PPMA pre-training strategy on a combination of no-human real data and synthetic data, we were able to achieve similar performance to models trained on real-human data. Compared to SynAPT, we made significant progress bridging the performance gap between representations trained on data with humans and without humans.

We summarize our contributions as follows:

1. We are the first to study the transferability of privacy-preserving representations learned from real-world videos without humans (by curating the No-Humans Kinetics dataset) to a variety of human action recognition datasets.

2. We propose a novel extension of the SynAPT benchmark [75], where the model is pre-trained on both *real videos without humans* and synthetic videos containing *virtual humans*, and then evaluated on a diverse set of downstream tasks.

3. To combine human-removed real data and synthetic videos, we propose a new pre-training strategy called Privacy-Preserving MAE-Align (PPMA). Compared to baselines, our approach improves downstream task performance by up to 5% on average.

## 2   Related Work

**Video Datasets for Pre-training.** Large-scale video datasets, such as YouTube 8M [1], Kinetics [42], and Moments-in-Time [54], are crucial for training large models. Pre-training on these datasets is especially important when the target dataset is limited in size. While traditional datasets focus on human-annotated action labels, recent efforts have explored alternative annotations like social-media videos [29], instructional narrations [53], and spoken captions [55]. However, using real video data with humans could raise privacy and ethical concerns. In this paper, we propose a novel approach to address these issues by pre-training on human-free real and synthetic video data.

**Privacy-Preserving Action Recognition.** Prior methods for privacy-preserving action recognition involve downsampling [21], blurring [11], using off-the-shelf detectors to blur specific regions like the face [57] [82], replacing faces with synthetic faces [60], or even using adversarial image modification to maximize privacy while preserving performance [58, 77, 22]. AViD [57] is a popular dataset that blurs faces, but complete privacy isn't guaranteed. Specifically, other features like height, clothing, or

body shape can still reveal individual identity [78, 56]. In our work, we eliminate humans entirely via mask prediction and inpainting. In addition, unlike these prior works, we study the transferability of privacy-preserving representations to a diverse set of downstream human action recognition datasets.

**Background Bias.** State-of-the-art action recognition models often overrely on background features such as objects and scene cues instead of foreground action content, which leads to sub-optimal performance when generalizing to other tasks [80, 18]. One solution to this problem is to add data augmentation that swaps the action and background of two videos [80, 84, 32]. Additionally, one can add an adversarial loss to reduce background bias [18]. Our proposed PPMA method is different in that we learn background features from real data and temporal information from synthetic data. Thus, when we generalize to new downstream tasks, the model has priors about background features but can also focus on temporal action features.

**Learning with Synthetic Data.** Synthetic data has been extensively researched as a substitute for real-world training data in computer vision [24, 59, 63]. It can serve as a cost-effective solution for scaling datasets and can also be utilized to preserve privacy by replacing humans with virtual counterparts in images or videos [6]. Previous works in action recognition have employed graphic engines to generate videos simulating real human actions [40, 68]. However, there exists a performance gap when using synthetic data to learn transferable representations compared to real videos [27]. SynAPT [75] discovered that synthetic action videos performed well on downstream tasks with low scene-object bias but performed worse on tasks with high scene-object bias. In contrast to previous approaches, our method combines real-world data without humans with synthetic data (virtual humans) for pre-training, resulting in significant improvements in representation transferability, particularly for tasks with high scene-object bias.

**Pre-training Transformers for Action Recognition.** The model architecture plays a crucial role in training high-performing action recognition models, whether using real or synthetic data. Vision transformers [25, 4] have emerged as the state-of-the-art, surpassing CNNs like I3D [15] and TSN [72] [31, 67, 30, 73]. Pre-training on large datasets is essential for the good predictive performance of these models [15, 28, 49, 26, 43]. While supervised pre-training can serve this purpose, it is limited by the cost of labeling large-scale data. Self-supervised pre-training serves as an effective replacement. Methods include contrastive learning [16, 38, 66], clustering [12, 13], self-distillation [14, 34], or masked auto-encoders (MAE) [37, 67]. In our work, we propose the Privacy-Preserving MAE-Align (PPMA) pre-training scheme. We first perform self-supervised pre-training with MAE and then proceed to supervised training for action recognition. This scheme provides us with more discriminative features than any of its individual components do. This observation is similar to findings of concurrent work [64].

# 3 Proposed Benchmark and Privacy Preserving MAE-Align (PPMA)

Our goal is to study the transferability of pre-trained representations on large scale human-free data that preserves privacy. Towards this end, we build on the SynAPT benchmark [75] by adding privacy-preserving real data (*i.e.*, No-Human Kinetics) (Sec. 3.1) for pre-training. We then pre-train state-of-the-art action recognition models using Privacy Preserving MAE-Align (PPMA) (Sec. 3.5) and evaluate them on 6 diverse downstream action recognition tasks (Sec. 3.3).

## 3.1 Privacy-Preserving Real Data

We aim to improve the transferability of pre-trained models by leveraging real data and remove privacy concerns. In this work, we choose Kinetics [42] as our real data. Kinetics is a large-scale video dataset consisting of 400 classes of human action clips curated from YouTube. To remove humans from these clips, in order to eliminate privacy and ethical issues with the data, we employ the HAT [19] pipeline. This consists of first using a SeMask segmentation model [41] to obtain per-frame segmentation masks around humans. The clips along with the segmentation masks are input to $E^2$FGVI [48], an optical flow-based inpainting model, to remove humans from the video. We run this pipeline on a subset of Kinetics training videos with 150 action classes and 1000 video clips per class following SynAPT. The outputs of these steps are videos without humans comprising the No-Human Kinetics (NH Kinetics) dataset. Fig. 2 shows some examples of these. In most cases, the HAT framework is effective at identifying and removing humans from all the frames in the video

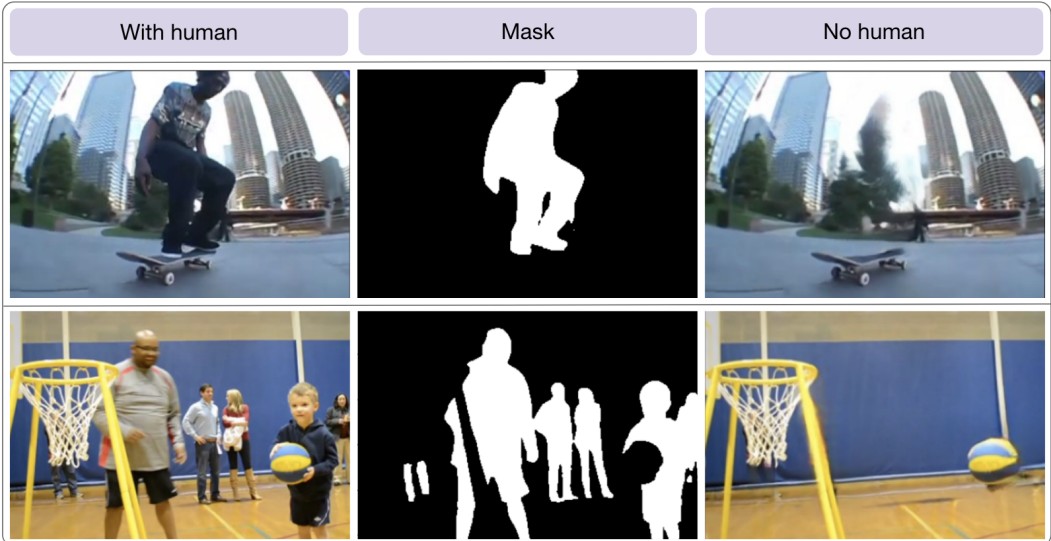

| With human | Mask | No human |

Figure 2: **Examples of privacy-preserving real video data without humans.** With the HAT [19] framework, we identify the human segmentation mask and remove the human from the video.

while keeping the context in place. As segmentation and inpainting are off-the-shelf models, we conduct a manual review of No-Human Kinetics and discuss limitations in Appendix B.

## 3.2 Synthetic Data

The pre-training data in our benchmark is also comprised of synthetic videos from SynAPT [75]. This includes videos of virtual humans performing various actions curated from ElderSim [40], SURREACT [68], and Procedural Human Action Videos (PHAV) [23]. Same as No-Human Kinetics, it has 150 action categories with 1000 examples each. One of the primary aspects of the synthetic data is that it de-correlates actions and backgrounds, helping the model focus on temporal features. This is a property often lacking in real videos where the actions may be correlated to the contexts they take place in. More information about assumptions and bias of this data is available in SynAPT [75] and its sources.

## 3.3 Downstream Evaluation

The pre-trained models' transferability is assessed using six diverse tasks from SynAPT [75] listed in decreasing order of scene-object bias: **UCF101** [65] features 13,320 YouTube videos across 101 action classes, offering significant variations in actions and camera motion. **HMDB51** [45] provides 6,849 clips, mostly from movies, divided into 51 action classes. **Mini Something-Something V2 (Mini-SSV2)** is a smaller 87-category version of the Something-Something V2 [33] dataset, emphasizing fine-grained human hand gestures with everyday objects. **Diving48** [47], a specialized dataset for competitive diving, tests a model's robustness due to its similar background and object features. It has 18,000 clips from 48 categories. **Ikea Furniture Assembly** [7] provides 111 videos of 14 actors assembling and disassembling furniture under consistent camera and scene settings. The dataset has 12 action categories. **UAV-Human** [46] contains 22,476 videos captured by unmanned aerial vehicles (UAVs), presenting 155 action classes and 119 subjects.

## 3.4 MAE-Align

While SynAPT [75] relied on models with a CNN architecture, as mentioned in the introduction, state-of-the-art action recognition models use vision transformers (ViT). However, these ViT models require pre-training on additional data such as Imagenet-21K, which contain additional pictures of humans, giving up privacy even when no humans appear in videos. Thus, we choose to leverage MAE [67] because it outperforms other state-of-the-art self-supervised method for action recognition

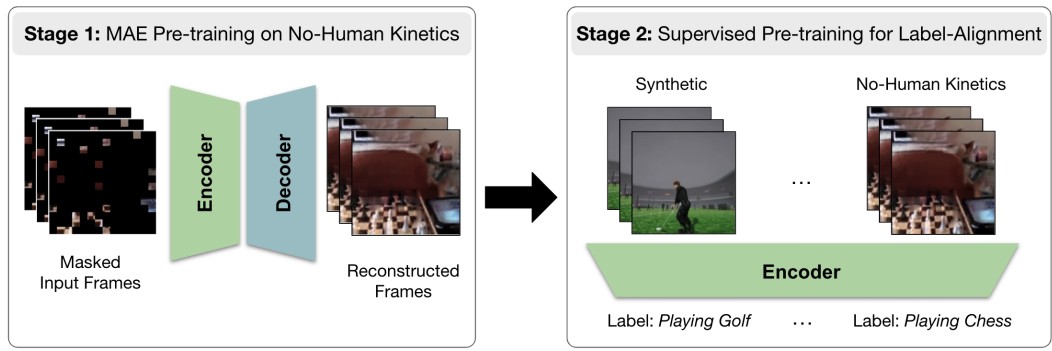

Figure 3: **Privacy-Preserving MAE-Align (PPMA). Stage 1:** An encoder-decoder transformer is pre-trained for the MAE reconstruction task with privacy-preserving (no human) real video data in order to learn the context features of actions. **Stage 2:** the encoder is further pre-trained on synthetic human action recognition datasets for alignment with the action recognition task.

[67, 71] and it is data-efficient, meaning we do not need to pre-train on additional image data such as Imagenet-21K. Specifically, we propose a two-stage pre-training approach:

**Stage 1: MAE Training.** We use the standard MAE training process for videos [67, 30]. An encoder-decoder architecture is trained to predict masked pixel values in videos, following which the decoder is discarded.

**Stage 2: Supervised Alignment.** To the trained MAE encoder, we add a linear classifier layer and train this whole network for classification with supervision from action labels.

The two stages are crucial for learning discriminative features for downstream tasks, as we shall show in Sec. 4.3.

### 3.5 Privacy-Preserving MAE-Align (PPMA)

Fig. 3 shows our approach. Using the no-human data from our benchmark, we present Privacy Preserving MAE-Align (PPMA) as a method of learning strong transferable video representations and maintaining privacy, while simultaneously closing the performance gap to pre-training with video data containing real humans (Kinetics) (see Tab. 1). Stage 1 involves MAE training with No-Human Kinetics. The reconstruction task allows features to learn the real-world context of actions. Stage 2 of PPMA involves supervised alignment using videos from No-Human Kinetics and SynAPT Synthetic data. As the size of the No-Human Kinetics and Synthetic dataset are roughly equal and we use the same loss function for both sources of data, both datasets have roughly equal impact on training the model. Additional training details can be found in Appendix A.

## 4 Experiments

### 4.1 Implementation Details

For all experiments, we use the ViT-B [4] model. We train all models from scratch without ImageNet pre-trained weights. First, we pre-train the model with MAE for 200 epochs using the random masking [67] strategy. After this, we discard the MAE decoder and only keep the encoder. We perform supervised pre-training for 50 epochs during the label-alignment stage. For comparisons using Kinetics data, we use the same 150 class subset as used in SynAPT [75]. For all downstream tasks, we finetune (FT) the entire network or train a linear probe (LP) for 30 epochs. More training details can be found in Appendix A.

### 4.2 PPMA Performance

Tab. 1 reports the individual task and average downstream performance of different pre-trained representations including our proposed Privacy Preserving MAE-Align (PPMA). The table is split into two halves with the top half containing representations that do not preserve privacy and the

Table 1: **PPMA performance.** Reported are top-1 downstream task accuracy for finetuning (FT) and linear probing (LP). Average represents the average FT and LP accuracy across all downstream tasks. Unless specified, a ViT-B backbone is used. We find that Privacy Preserving MAE-Align (PPMA) is 2.5% better with FT and 5% better with LP than the next best baseline (MAE-Align w/ Synthetic). It learns features comparable to pre-training with data with real humans. NH = No-Human.*uses an Imagenet-21K pre-trained backbone before training with videos.†results reported in [75].

| Pre-trained Model | Privacy Preserving | Stage 1: MAE | Stage 2: Alignment | UCF101 | | HMDB51 | | Mini-SSV2 | | Diving48 | | IkeaFA | | UAV-Human | | Average | |
|---|---|---|---|---|---|---|---|---|---|---|---|---|---|---|---|---|---|
| | | | | FT | LP | FT | LP | FT | LP | FT | LP | FT | LP | FT | LP | FT | LP |
| MAE-Align w/ real humans | ✗ | Kinetics | Kinetics | **93.3** | **91.4** | **73.4** | **69.5** | **68.8** | 37.4 | **66.3** | 19.9 | **72.0** | 58.3 | 34.9 | 13.9 | **68.1** | **48.4** |
| TimeSformer Kinetics*† | ✗ | x | Kinetics | 92.1 | 89.4 | 59.5 | 55.4 | 48.9 | 21.5 | 46.4 | 17.0 | 61.9 | 47.7 | 23.3 | 8.4 | 55.3 | 39.9 |
| TimeSformer Synthetic*† | ✗ | x | Synthetic | 89.0 | 82.1 | 54.4 | 49.2 | 51.1 | 21.2 | 44.9 | 19.2 | 63.6 | 45.5 | 25.0 | 13.8 | 54.7 | 38.5 |
| Scratch | ✔ | x | x | 30.1 | - | 14.8 | - | 16.0 | - | 9.3 | - | 19.5 | - | 0.7 | - | 15.1 | - |
| TSN (RN50 backbone)† | ✔ | x | Synthetic | 83.4 | 28.0 | 54.4 | 20.9 | 49.7 | 12.8 | 63.5 | 10.9 | 42.7 | 36.0 | 35.6 | 5.7 | 54.9 | 19.1 |
| I3D (RN50 backbone)† | ✔ | x | Synthetic | 82.1 | 27.6 | 55.7 | 22.6 | 50.7 | 12.3 | 55.3 | 10.1 | 42.7 | 33.2 | 35.1 | 5.8 | 53.6 | 18.6 |
| R(2+1)D (RN50 backbone)† | ✔ | x | Synthetic | 80.0 | 26.4 | 53.3 | 22.2 | 52.0 | 13.3 | 57.3 | 10.0 | 41.5 | 35.7 | 31.8 | 5.5 | 52.6 | 18.9 |
| MAE-Align w/ Synthetic | ✔ | Synthetic | Synthetic | 88.7 | 76.6 | 69.7 | 59.7 | 64.3 | 26.0 | 61.1 | 16.7 | 67.3 | **57.7** | 36.1 | **20.6** | 64.5 | 42.9 |
| **Ours : Privacy-Preserving MAE-Align (PPMA)** | ✔ | NH Kinetics | NH Kinetics + Synthetic | 92.5 | 88.4 | 71.2 | 64.9 | 67.8 | 34.9 | 64.0 | 21.9 | 67.9 | 57.7 | **38.5** | 19.3 | 67.0 | 47.9 |

bottom, representations that do. These models pre-trained on conventional large-scale real video data with humans have a performance edge over models trained with synthetic data due to a realism gap. The best performing model with real human videos is "MAE-Align w/ real humans" in Tab. 1.

While other baselines (described below) which preserve privacy present a significant performance gap, with PPMA and the use of No-Human Kinetics, the average downstream performance gap from the human-baseline is down to 1.1% (68.1% vs 67%) with FT and 0.5%. One reason for this gap is that PPMA does worse than the human-baseline on high scene-object bias tasks such as UCF101, HMDB51, and Mini-SSV2. This could be because including humans in the videos of Kinetics helps the model better learn both scene-object cues and action features. Nevertheless, compared to training solely with synthetic data in the MAE-Align w/ Synthetic model, PPMA has significantly reduced the gap with the human-baseline. Thus, while preserving privacy, we can achieve comparably good pre-trained representations for action recognition.

"Scratch" refers to randomly initializing a ViT-B backbone and training it with label supervision on downstream tasks. Comparing its performance with the other methods offers an understanding of the downstream performance benefit of pre-training representations.

"TSN (RN50 backbone)" is the best model trained entirely using synthetic data from SynAPT [75]. This model lags "MAE-Align w/ real humans" significantly in downstream performance. A contributing factor to this is the limitation in the backbone architecture. Using the same data to pre-train a high capacity ViT-B backbone requires a more data efficient pre-training process which we achieve with MAE-Align (see further discussion of data efficiency in 4.3). This pre-trained model, "MAE-Align w/ Synthetic" indeed improves average downstream performance significantly (from 54.9% to 64.5% FT and from 19.1% to 42.9% LP). Also noteworthy is the fact that using MAE-Align and synthetic data, this model outperforms both ViT-B architecture based TimeSformer models from [75], indicating the strength of MAE training compared to large scale supervised pre-training on Imagenet-21K. Finally when we add No-Human Kinetics data to the mix, PPMA further improves over "MAE-Align w/ Synthetic" by 2.5% FT and 5% LP.

## 4.3 The effectiveness of MAE-Align

To understand the effectiveness of our pre-training strategy, using each of the three datasets: Kinetics, Synthetic, and NH Kinetics, we ablate one of the pre-training stages. The average fine-tune downstream accuracy over the 6 tasks, is reported in Tab. 2. Fig. 4 shows results for these pre-trained models with FT for different downstream tasks. The same analysis with LP is in Appendix C.

Comparing only MAE pre-training to only supervised alignment, we find features are more transferable (with higher average downstream task performance) in the former case than the latter. This

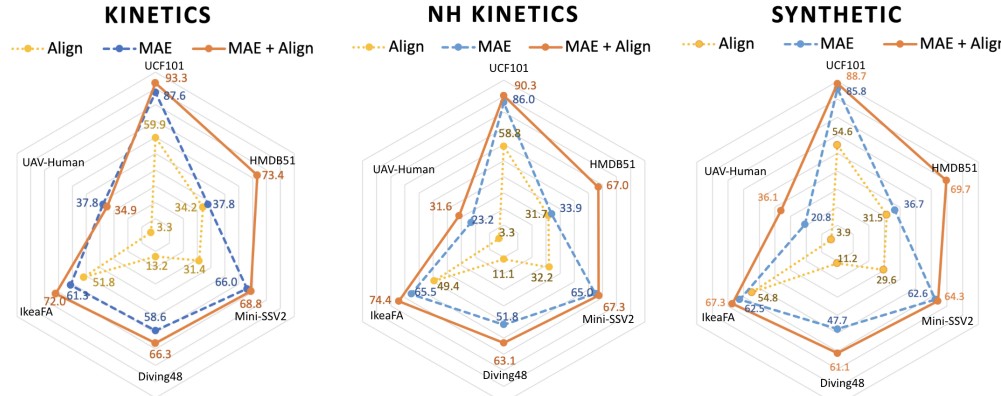

Figure 4: **Ablating stages 1 and 2 of MAE-Align for different pre-training datasets.** The title of each chart indicates the pre-training dataset. Each chart plots the downstream accuracies of the models finetuned for the 6 downstream tasks. We show three different model performances for each downstream task with three different pre-training strategies: Align only, MAE only, and MAE+Align. Our results show that both MAE and Align stages are crucial for good downstream performance.

is because high-capacity models like ViTs with a classification objective have a loss landscape with several sharp local minima, making it hard to optimize using fewer data without large-scale pre-training (*e.g.* with ImageNet-21K) [17]. The MAE objective is more data-efficient in learning generalizable representations for these high-capacity networks [67].

While MAE pre-training helps with features when finetuned on downstream tasks, the improvement compared to supervised training only is smaller when only a linear probe is learned. We hypothesize that the MAE pre-trained backbone is not well aligned with the downstream action recognition tasks. Hence, when the features are frozen (LP), the performance of the downstream classifier is not significantly higher, like in the case of FT. On training with supervised labels on the pre-training set after MAE, the backbone is more aligned to action recognition and improves performance significantly even with LP.

## 4.4 Combining Contextual and Temporal Features

Pre-training on synthetic data is helpful for learning temporal action features but not contextual features [75]. This is compensated for by using no-human real data to learn contextual features.

In Tab. 2, we find that MAE pre-training on No-human Kinetics learns more transferable features than MAE on Synthetic data. The reconstruction objective of MAE helps models learn the contexts of actions making it more suitable for videos with real backgrounds.

After MAE pre-training on No-Human Kinetics, we explore the effect of supervised alignment on different datasets: Synthetic, No-Human Kinetics or a combination of the two. In Tab. 3 we find that on average over the 6 downstream tasks, combining label alignment using both these datasets as we propose in PPMA, leads to the most transferable features.

Table 2: **Ablating stages 1 and 2 of MAE-Align for different pre-training datasets.** Average downstream accuracy over 6 tasks is reported. Each row also mentions the performance difference to performing only supervised alignment under $\Delta$FT and $\Delta$LP. We find that MAE is better for pre-training features than supervised alignment but a subsequent alignment stage improves representations even further.

| Stage 1 : MAE | Stage 2 : Alignment | Average Downstream | | | |
|---|---|---|---|---|---|
| | | FT | $\Delta$FT | LP | $\Delta$LP |
| x | Kinetics | 32.3 | - | 25.1 | - |
| Kinetics | x | 59.9 | +27.6 | 28.7 | +3.6 |
| Kinetics | Kinetics | 68.1 | +35.8 | 48.4 | +23.3 |
| x | NH Kinetics | 31.1 | - | 25.3 | - |
| NH Kinetics | x | 58.0 | +26.9 | 28.4 | +3.2 |
| NH Kinetics | NH Kinetics | 65.6 | +34.5 | 42.4 | +17.2 |
| x | Synthetic | 31.1 | - | 25.4 | - |
| Synthetic | x | 56.4 | +25.3 | 27.8 | +2.4 |
| Synthetic | Synthetic | 64.5 | +33.4 | 42.9 | +17.5 |

**Averaging Model Weights.** Among alternate ways of combining temporal (from Synthetic data) and contextual (from No-Human Kinetics) features, we additionally explored averaging pre-trained model weights as done in [76]. Tab. 4 contains the results of averaging the weights of the three pre-trained models from Tab. 3 in the proportions $\alpha_1$, $\alpha_2$ and $\alpha_3$ respectively, where $\alpha_1 + \alpha_2 + \alpha_3 = 1$. In the first row of Tab. 4 we combine two models which are aligned using NH-Kinetics and Synthetic respectively, in equal proportion. In contrast to PPMA, which label aligns one model on NH Kinetics + Synthetic, this separately label-aligns two models on the two datasets and later combines the trained model weights. In performance, we find that this lags behind PPMA, thus indicating label-alignment on combined data is the better strategy for learning more transferable features.

Next, we evaluate averaging the weights of three models adding PPMA to the mix above. Row 2 of Tab. 4 shows the results of mixing the three models in equal proportion and row 3, the results of mixing them in proportion of the amount of data used for label-alignment. We find that these mixtures outperform the mixture of two models, with the last mixture performing significantly better. These results indicate averaging models pre-trained on different subsets of our human-free data could be effective representations for downstream tasks. We leave further exploration of this to future work.

Table 3: **Supervised alignment with temporal and contextual features.** Comparing the downstream performance of models label-aligned on No-Human (NH) Kinetics (contextual), Synthetic (temporal), or both datasets (contextual and temporal). All models were first pre-trained with MAE on No-Human Kinetics. Note that the first row is our proposed PPMA, which performs better downstream compared to label-alignment using either of the individual datasets, Synthetic or NH Kinetics.

| Stage 1 : MAE | Stage 2 : Alignment | Downstream Task Accuracy | | | | | | | | | | | | Average | |
|---|---|---|---|---|---|---|---|---|---|---|---|---|---|---|---|
| | | UCF101 | | HMDB51 | | Mini-SSV2 | | Diving48 | | IkeaFA | | UAV-Human | | | |
| | | FT | LP | FT | LP | FT | LP | FT | LP | FT | LP | FT | LP | FT | LP |
| NH Kinetics | NH Kinetics | 90.3 | 84.5 | 67.0 | 58.1 | 67.3 | 32.5 | 63.1 | 17.8 | **74.4** | 51.2 | 31.6 | 10.6 | 65.6 | 42.4 |
| | Synthetic | 91.4 | 81.9 | **71.5** | 62.0 | 66.4 | 31.8 | **65.3** | 21.8 | 67.3 | **57.7** | 38.3 | **20.8** | 66.7 | 46.0 |
| | NH Kinetics + Synthetic | **92.5** | **88.4** | 71.2 | **64.9** | **67.8** | **34.9** | 64.0 | **21.9** | 67.9 | **57.7** | **38.5** | 19.3 | **67.0** | **47.9** |

Table 4: **Averaging model weights to combine temporal and contextual features.** The three models label-aligned on No-Human Kinetics, Synthetic, and both datasets are combined in proportions of $\alpha_1$, $\alpha_2$ and $\alpha_3$ respectively. We find that averaging the first two models in equal proportion performs slightly poorer than PPMA. Adding PPMA into the mix with a non-zero $\alpha_3$ improves downstream performance significantly.

| Model Proportions | | | Downstream Task Accuracy | | | | | | | | | | | | Average | |
|---|---|---|---|---|---|---|---|---|---|---|---|---|---|---|---|---|
| NH Kinetics $\alpha_1$ | Synthetic $\alpha_2$ | Both $\alpha_3$ | UCF101 | | HMDB51 | | Mini-SSV2 | | Diving48 | | IkeaFA | | UAV-Human | | | |
| | | | FT | LP | FT | LP | FT | LP | FT | LP | FT | LP | FT | LP | FT | LP |
| 0.50 | 0.50 | 0.00 | 91.6 | 85.0 | 70.8 | 63.4 | 67.2 | 36.8 | 67.5 | 21.9 | 65.2 | **63.4** | 36.9 | 15.8 | 66.5 | 47.7 |
| 0.33 | 0.33 | 0.33 | 92.2 | 85.5 | **72.7** | 64.1 | **67.8** | 37.3 | 68.0 | 22.1 | 70.1 | 59.8 | 38.0 | 17.1 | 68.1 | 47.6 |
| 0.25 | 0.25 | 0.50 | **93.0** | **86.6** | 72.1 | **65.9** | **67.8** | **37.7** | **69.3** | **23.2** | **73.2** | 61.6 | **38.6** | **18.6** | **69.0** | **48.9** |

## 4.5 Averaging Model Weights with Learned Proportions

In Tab. 5, on changing the proportions $\alpha_1$ and $\alpha_2$ of combining model weights, we find different ratios of alignment on No-Human Kinetics and Synthetic can improve the representations for downstream tasks. This motivates exploring a method where we can learn, based on the downstream task, a proportion of mixing the two models for optimal performance. The last row of Tab. 5 reports performance of this model, where $\alpha_1 = 1 - \beta$ and $\alpha_2 = \beta$. $\beta$ is a parameter initialized to $1/2$ and learned via gradient descent during LP (while the pre-trained representations are frozen and only the linear probe is updated). In practice the constraint $\beta \in [0, 1]$ is enforced by making it the output of a softmax function. Note that we only learn this mixing proportion of pre-trained weights for LP, since this mode of evaluation keeps those weights frozen. This would not be possible if the backbone weights are updated while learning the parameter as is the case in FT.

From the results reported in the last row of Tab. 5, we observe that learning this mixing proportion per dataset benefits the performance over PPMA or the pre-defined proportions we evaluated on Ikea-FA and UAV-Human datasets. On the rest, it is on par with (UCF101, Diving48) or worse than (HMDB51, Mini-SSV2) the best mixing proportion. This is possibly due to $\beta$ fitting aggressively to the training set of downstream tasks. Tab. 6 presents the learned $\beta$ at the end of training for

Table 5: **Adaptive averaging model weights to combine temporal and contextual features.** Models are label-aligned on No-Human Kinetics and Synthetic and averaged in proportions of $\alpha_1$ and $\alpha_2$ respectively. For the bottom row, we learn adaptive mixing ratio $\beta$ for each downstream task with LP.

| Model Proportions | | Downstream Task Accuracy | | | | | | | | | | | | Average | |
|---|---|---|---|---|---|---|---|---|---|---|---|---|---|---|---|
| $\alpha_1$ | $\alpha_2$ | UCF101 | | HMDB51 | | Mini-SSV2 | | Diving48 | | IkeaFA | | UAV-Human | | | |
| | | FT | LP | FT | LP | FT | LP | FT | LP | FT | LP | FT | LP | FT | LP |
| PPMA (Mixing data) | | **92.5** | **88.4** | 71.2 | **64.9** | **67.8** | 34.9 | 64.0 | **21.9** | 67.9 | 57.7 | **38.5** | 19.3 | 67.0 | 47.9 |
| 0.25 | 0.75 | 91.5 | 82.9 | **72.1** | 64.8 | 66.6 | 34.8 | 66.4 | 21.8 | 67.9 | 64.0 | 37.8 | 20.2 | 67.0 | **48.1** |
| 0.5 | 0.5 | 91.6 | 85.0 | 70.8 | 63.4 | 67.2 | **36.8** | **67.5** | **21.9** | 65.2 | 63.4 | 36.9 | 15.8 | 66.5 | 47.7 |
| 0.75 | 0.25 | 91.8 | 86.0 | 69.3 | 59.7 | 67.7 | 36.2 | 66.1 | 21.3 | **73.8** | 60.4 | 34.5 | 12.8 | **67.2** | 46.1 |
| $1-\beta$ | $\beta$ | - | 85.0 | - | 60.3 | - | 34.7 | - | 21.6 | - | **64.6** | - | **20.4** | - | 47.8 |

Table 6: **Learned mixing ratio for different downstream tasks.** We adaptively learn the mixing ratio of Synthetic and No-Human Kinetics models for each downstream task.

| | UCF101 | HMDB51 | Mini-SSV2 | Diving48 | IkeaFA | UAV-Human |
|---|---|---|---|---|---|---|
| Percentage of Synthetic: $\beta$ | 14.3% | 21.5% | 66.0% | 75.8% | 71.9% | 83.0% |
| Percentage of Real: $1-\beta$ | 85.7% | 78.5% | 34.0% | 24.2% | 28.1% | 17.0% |

each of the downstream tasks and we see a preference for a higher ratio for the Synthetic backbone in tasks with low scene-object bias (Mini-SSV2, Diving48, IkeaFA, UAV-Human). This might be expected since real data helps the model learn contextual cues whereas synthetic data helps the model learn temporal action features. We leave further investigation of this model and other techniques of combining pre-trained representations to future work.

## 5   Conclusion

While real world videos make for excellent pre-training data for action recognition models, they are riddled with privacy issues. Prior use of synthetic data to circumvent these issues was limited in performance because of no exposure to real-world *context* of actions. In this work, we propose a benchmark to overcome these challenges by using real videos with obfuscated humans, along with synthetic videos for pre-training. Using this data we proposed Privacy-Preserving MAE-Align (PPMA), a method for pre-training representations that while preserving privacy, learns features that are at par with pre-trained representations on real human data in terms of transferability to downstream action recognition tasks.

**Limitations.** While PPMA is effective, additional representation learning methods could improve upon it. One potential direction of exploration is using averages of multiple pre-trained models, that we leave for future work. Also beyond the scope of our current work is a study of how privacy-preserving pre-training methods scale with more videos, presenting an interesting direction for future research.

**Potential Social Impacts.**   Our work aims to mitigate the negative impacts of using data with real humans by obfuscating them and preserving their privacy. Additionally, this can make models pre-trained on this data less biased towards aspects such as races or genders of humans appearing in the data. Potential negative impacts of this work include simply applications where automated action recognition can be used to make rash decisions without human intervention. One example of this could be automated identification of people and incrimination of activities deemed illicit in camera feeds.

**Acknowledgement.** This material is based upon work supported by the Defense Advanced Research Projects Agency (DARPA) under Contract No. FA8750-19-C-1001. Any opinions, findings and conclusions or recommendations expressed in this material are those of the author(s) and do not necessarily reflect the views of the Defense Advanced Research Projects Agency (DARPA). This work was partially supported by Institute of Information and Communications Technology Planning and Evaluation (IITP) grant funded by the Korea government (MSIT) (No. 2019-0-00079, Artificial Intelligence Graduate School Program, Korea University).

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

# Appendix

## A  Training Details

Our implementation is based upon OmniMAE [30] repository. All models were trained distributed over 4-8 NVIDIA V100 and A100 GPUs.

### A.1  Pre-training

#### A.1.1  Stage 1: MAE

**Architecture**: All experiments use the ViT-B backbone architecture. The encoder consists of 12 transformer blocks with embedding dimension of 768, and the decoder consists of 4 transformer blocks with an embedding dimension of 384.

**Input:** We represent the video as a 4D tensor of shape $T \times H \times W \times 3$ where $T$ represents the temporal dimension, $H$ and $W$ are spatial dimensions, and 3 is the number of color channels. We then convert this 4D tensor into $N$ spatio-temporal patches, each of size $t \times h \times w \times 3$. We set $t = 2$ and $h = w = 16$.

**Loss Function:** To minimize the reconstruction error between the image pixel values and the decoder's prediction, we measure reconstruction error with $\ell_2$ loss after normalizing the pixels to zero mean and unit variance.

**Masking Ratio:** For each frame, we mask out 90% of the patches in the image at random. The high masking ratio is chosen to make the reconstruction task more difficult and encourage extracting more effective video representations. While VideoMAE [67] advocates to use tube masking for small improvements in accuracy, we decide to follow the training details in OmniMAE and use the random masking scheme [30].

**Other Training Details:** We train our model with a mini-batch size of 128. Each batch consists of 32 distinct videos, each replicated 4 times but with different masks as in [30]. We resize the videos spatially to $224 \times 224$ pixels and sample a clip of $T = 16$ frames. The model takes in patch sizes of $2 \times 16 \times 16$ and uses a non-learnable sinusoidal positional encoding. We choose to use global pooling instead of a class token. We use the AdamW optimizer [52, 44] with a cosine learning rate scheduler [51] that uses a maximum learning rate of 8e-4. We train for 200 epochs, with a 10 epoch learning rate warmup that linearly grows from 5e-7 to 8e-4.

#### A.1.2  Step 2: Supervised Alignment

After MAE pre-training, we discard the decoder and add a linear classification layer to the encoder to perform action recognition on No-Human Kinetics and synthetic data. This is a crucial step to align the model's features learned from MAE to downstream action recognition tasks.

**Architecture**: We use the same ViT-B encoder and add a final linear classification layer for supervised training. When co-training on two datasets, the model shares an encoder but has a different linear classification layer for each dataset.

**Input:** This input is the same as in MAE as described in Appendix A.1.1.

**Loss Function:** We minimize cross entropy loss between the final classification layer output and the true action category.

**Details for training on two datasets:** When training on both No-Human Kinetics and Synthetic, each batch consists of samples from one dataset: either only real or only synthetic data. Both sources of data are roughly equal size, and each epoch involves training on every example of No-Human Kinetics and Synthetic data once. Loss functions for real and synthetic data are the same and are weighted equally.

**Other Training Details:** Most training parameters, such as the mini-batch size of 128, are the same as in Appendix A.1.1. However, we train for only 50 epochs and choose different hyperparameters for our optimizer. The cosine rate scheduler we use has a maximum learning rate of 2e-3, with a 6 epoch learning rate warmup where learning rate linearly grows from 5e-7 to 2e-3. We add data

augmentation techniques such as RandAugment [20], Random Erasing [83], Mixup [81], and CutMix [79].

## A.2 Downstream Evaluation Description

We evaluate the pretrained action recognition representations on six diverse, downstream tasks described in Sec. 3.3 of the main paper. Starting from the pre-trained ViT-B encoder, for the specific downstream task, we add a linear classification layer. We train for 30 epochs with a mini-batch size 32. In the finetuning (FT) setting, we employ the same number of warmup steps, same data augmentations, and same maximum learning rate of 2e-3 for the cosine rate scheduler as in supervised alignment (Appendix A.1.2). For the linear probing stage, we only tune the final linear classification layer and use the same hyperparameter settings as FT except the cosine rate scheduler has a maximum learning rate of 1.6e-2.

# B Quality Assessment of No-Human Kinetics

In this section, we present the findings from a manual quality assessment of our inpainting pipeline to remove humans. 4 random samples from each of 150 categories were reviewed by 3 different people. We found that in 96.3% of the videos humans were completely removed. Additionally, on a subsequent review of the remaining 3.7% of the videos, we found in all of them, either the humans were too small or mixed in a crowd and hence unidentifiable on visually inspecting the inpainted video. We display more qualitative results/visuals for both failure and correct cases.

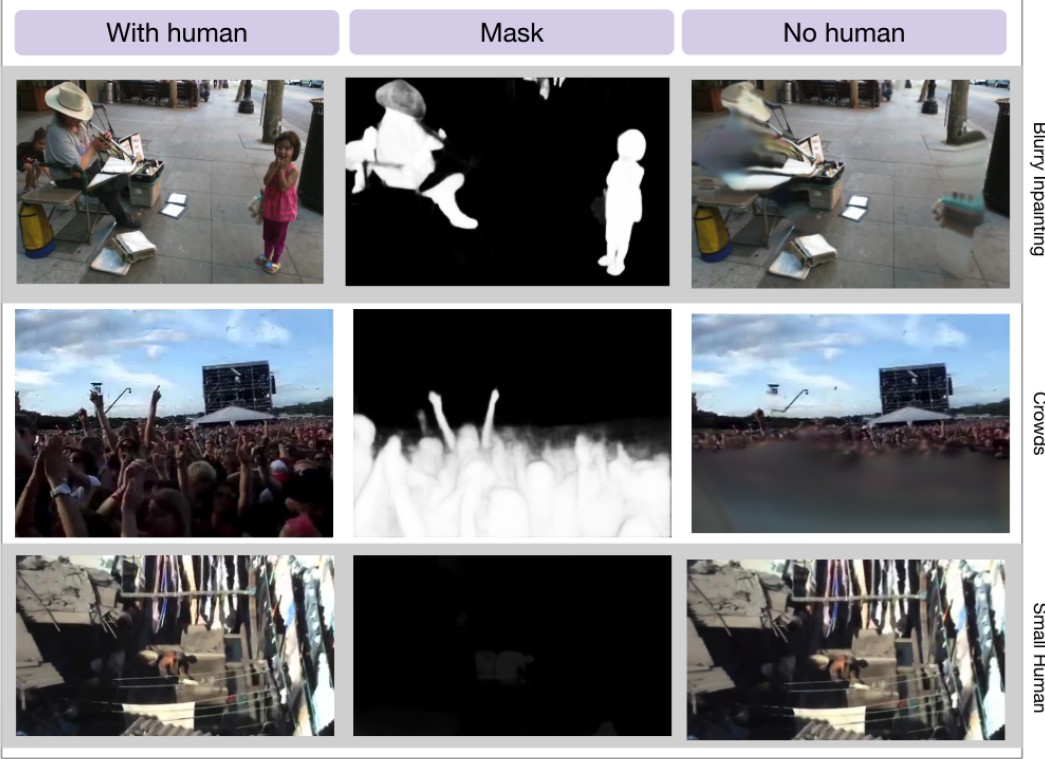

Figure 5: **Limitations of the HAT inpainting pipeline.** Displayed are examples of failure cases. In the top example, the human is connected to hat and trump and then inpainting becomes blurry. The 2nd and 3rd example showcase the error where not all humans are removed, which is more likely to occur when there are crowds of people or the human is small. In these cases, the segmentation model did not detect all humans in the video.

Fig. 5 shows examples of failure cases. The HAT inpainting pipeline can fail to accurately remove all humans from videos (i) with crowds (row 2) and (ii) when the human is too small (rows 2 and

3). Often, the errors result from the segmentation model not accurately detecting the human. In addition, the inpainted region may sometimes blurry (row 1), which may occur if the human is touching different objects or if there is poor lighting. While the human is still unidentifiable, this could be source of noise for representation learning.

Nevertheless, as seen from the manual review, the majority of inpaintings are able to remove humans and are effective for representation learning. See Fig. 6 for additional examples of where the segmentation network identifies the human and thus is able remove them.

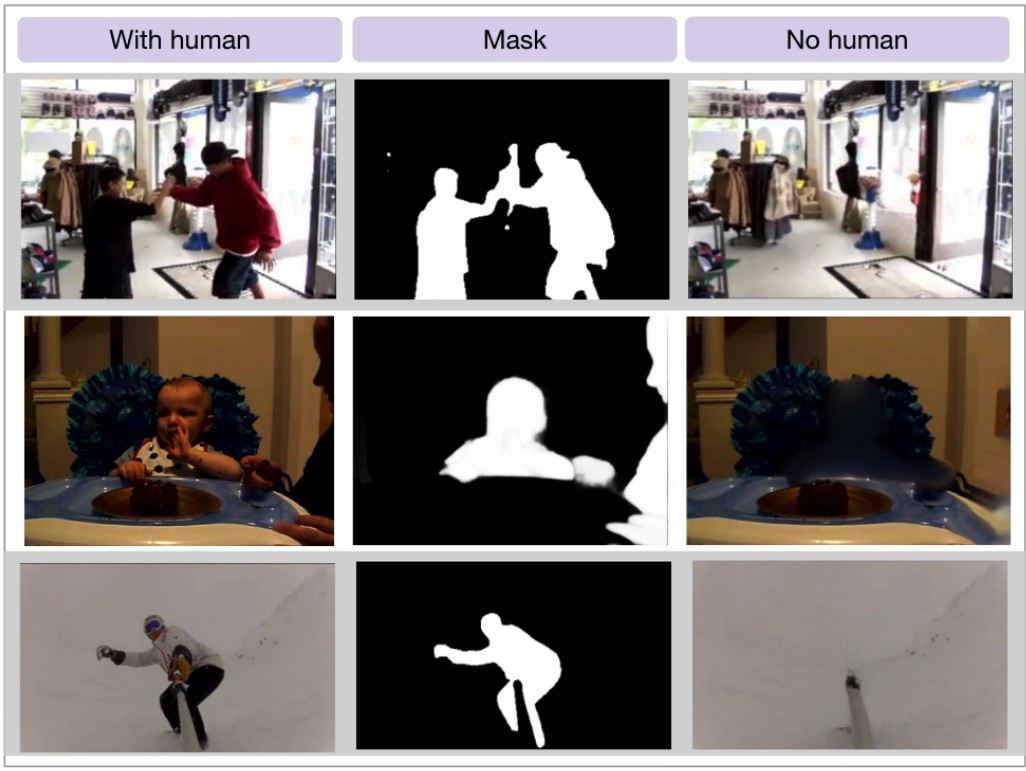

Figure 6: **Additional examples of No-Human Kinetics.** For the most part, HAT successfully removes the human from the video.

While Fig. 1 and Fig. 2 show examples where HAT successfully removed the human, we display failure cases where the segmentation model or inpainting model fails to identify all humans in Fig. 5. For example, at the top scene of Fig. 5, even after applying HAT to remove the humans, the video still shows remnants of spectators in the crowd and basketball players who are moving. We found that these errors of not removing all humans occur only in a few edge cases (3.7% total, as mentioned above): (i) videos with crowds (top 2 rows); (ii) when the human is moving too fast and is blurry and non-identifiable (top row); (iii) if the human is too small and non-identifiable (3rd row); (iv) or if the face was blurry and non-recognizeable to begin with (4th row). We believe that the issue mainly arises when the off-the-shelf segmentation network fails to detect the human, and further improvements in these aspects are possible with additional training of this network.

Additionally, another failure case of human removal is that inpainting network sometimes does not fully remove the outline of the human. Thus, while it is impossible identify the person, one may be able to infer information such as body shape or age. As segmentation and inpainting networks improve, we expect these issues to occur even less.

## C   MAE Align for Linear Probing

Analogous to Fig. 4 from the main paper, where downstream performance is evaluated with downstream finetuning(FT), we report linear probe(LP) downstream performance in Fig. 7. With LP, same

as with FT, we find MAE+Align for pre-training performs better than either MAE or Align. In Sec 4.3 of the main paper, we discuss MAE pre-training is not well aligned to downstream action recognition using average downstream performance (reported in Tab. 2). This results in poor downstream performance when the backbone features are frozen during LP. The same effect can be observed for each individual downstream task in Fig. 7.

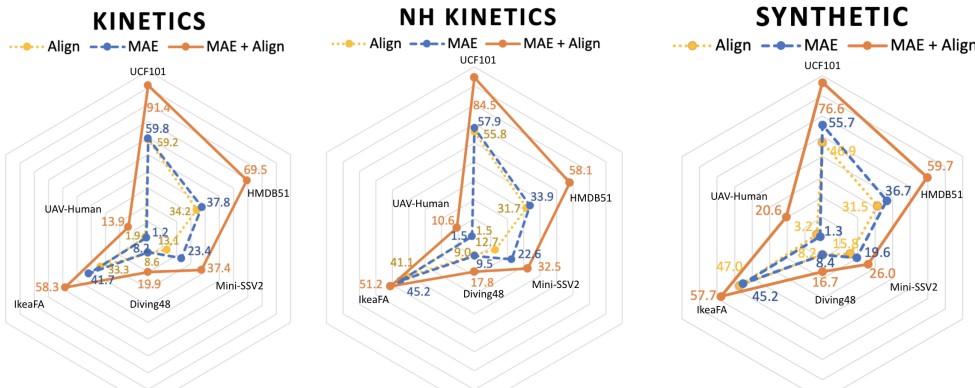

Figure 7: **Ablating stages 1 and 2 of MAE-Align for different pre-training datasets (LP).** The title of each chart indicates the pre-training dataset. Each chart plots the downstream accuracies of the models linear probed for the 6 downstream tasks. We show three different model performances for each downstream task with three different pre-training strategies: Align only, MAE only, and MAE+Align. Our results show that both MAE and Align stages are crucial for good downstream performance.

