# OpenReview forum: "Learning Human Action Recognition Representations Without Real Humans"
_NeurIPS.cc/2023/Track/Datasets_and_Benchmarks — NeurIPS 2023 Datasets and Benchmarks Poster_

### Official Review · Reviewer_He2J · 2023-07-17
**Review of PPMA**

**Rating:** 6
**Confidence:** 4
**Correctness:** Need more analysis. Details are given…
**Clarity:** Well.

**Strengths:**

+ The proposed method is interesting to use a mixture of synthetic data and processed real video for privacy protection.

+ The results seem good to show the advantage of the proposed method.

**Additional Feedback:**

What is the real and syn data mixture ratio in batch sampling and training? What are the weights of the loss of the real and syn data?

Tab. 4: please add the model names in the ratio places.

**Documentation:**

Yes.

**Limitations:**

- The data contribution is a little marginal, with only the real data processing and without manual checking.

- More details of the models and pre-training should be given to determine the factors affecting the performance.


**Opportunities For Improvement:**

- More analyses should be added to support the claim entirely. For example, how to ensure the segmentation and inpainting quality in the first step? What is the influence of artificial inpaintings?

**Relation To Prior Work:**

Well.

**Summary And Contributions:**

This paper proposes a way of pretraining the action recognition model with the no-human videos and then fine-tuning it with the mixture of no-human and synthetic human action videos, to avoid privacy problems. Based on the discovery of SynAPT, this work goes further to combine the advantage of the background knowledge from the real videos and the privacy keeping of the synthetic data. Following the setting of previous works, the proposed method is evaluated on several widely-used benchmarks.

---

> ### Author Response · Authors · 2023-08-22
>
> *More analyses should be added to support the claim entirely. For example, how to ensure the segmentation and inpainting quality in the first step? What is the influence of artificial inpaintings?*
>
> Thank you for the suggestion. We conducted a manual review of a 600 video sample of inpaintings across three different people to ensure segmentation and inpainting quality. We found that in only 3.7% of the videos humans were not removed completely. However, even in these cases, they were typically small and often mixed in a crowd, making them unidentifiable by looking at the image.
>
> *The data contribution is a little marginal, with only the real data processing and without manual checking.*
>
> While we did not collect new data, we would like to emphasize that by processing off-the-shelf data, we created the PPMA benchmark for action recognition pretraining on data without humans. By establishing this benchmark, we establish the groundwork for future use to evaluate privacy-oriented action recognition. In addition, we manually checked the inpaintings (see user study above).
>
> *More details of the models and pre-training should be given to determine the factors affecting the performance.*
>
> In Appendix A, we describe the implementation details for each step: MAE (A.1.1), Align (A.1.2), and downstream evaluation (A.2). We detail batch size, model dimensions, masking ratio, input dimension, data augmentation, and optimizer hyperparameters. As mentioned in Appendix A, these were the same as the default values in OmniMAE [1]. Please let us know if there is any additional specific information that we can include to make this paper stronger.
>
> *What is the real and syn data mixture ratio in batch sampling and training? What are the weights of the loss of the real and syn data*
>
> We will add these details in the final version. The ratio of the size of the real dataset to the size of syn dataset is approximately 1 to 1. During training, each batch consists of samples from one dataset: either only real or only syn. Loss functions for real and syn data are the same and are weighted equally.
>
> *Tab. 4: please add the model names in the ratio places.*
>
> Thank you for this concrete suggestion. We will include it in the final version.
>
> [1] Girdhar, Rohit, et al. "Omnimae: Single model masked pretraining on images and videos." Proceedings of the IEEE/CVF Conference on Computer Vision and Pattern Recognition. 2023.

---

> > ### Comment · Reviewer_He2J · 2023-08-22
> > **Post-rebuttal**
> >
> > Thanks for the response. For the segmentation and inpainting error, analysis and discussion are essential as we may enlarge the dataset and use the same way to aug data. If the asked content is added, I think my concerns are mainly addressed.

---

> > > ### Author Response · Authors · 2023-08-24
> > >
> > > Thank you for the prompt reply. We are happy to include our analysis based on the manual checking of 600 inpainted videos, as well as more discussion, to the main paper. We would like to point out that our main goal is to make humans unidentifiable, which is achieved by our method, as shown in our manual verification. Specifically, we do not require a precise reconstruction of inpainted objects or precise dilated segmentation masks, as long as humans are not identifiable. That said, we do agree with the reviewer that an analysis of segmentation and inpainting errors would be a valuable addition to the paper. Given our segmentation and  inpainting methodology, it is possible to finely vary errors in each by perturbing the segmentation masks or adding noise to inpainting. An extreme case is to add 100% noise to the segmentation mask, although that would reveal features such as body shape and height, which is not desirable for privacy reasons. We are in the process of doing this analysis, but given that a new round of pretraining would take more time than the discussion period, we would only be able to include the results in the final version, in the case the paper is accepted.

---

### Official Review · Reviewer_Jm2h · 2023-07-21

**Rating:** 7
**Confidence:** 4
**Correctness:** They seem correct to me.
**Clarity:** The paper is very well written.

**Strengths:**

The idea of deleting the human in the video is very interesting.

The proposed two-stage training strategy is able to achieve the compared results with the model trained with human information.


**Additional Feedback:**

N.A.

**Documentation:**

Yes, the GitHub page seems to be well-documented.

**Ethics:**

None that I know of.

**Limitations:**

The reason why using the MAE is not introduced. As there are other self-supervised strategies, the motivation should be clear.

**Opportunities For Improvement:**

Currently, the proposed method only uses the ViT-B model as the backbone. The authors are suggested adding more backbones.






**Relation To Prior Work:**

Its relation to prior works is clearly discussed.

**Summary And Contributions:**

In the submission, the authors introduce new privacy-preserving representations, which are learned from real-world video without humans.
The authors also propose a new pre-training strategy, Privacy-Preserving MAE-Align (PPMA).

---

> ### Author Response · Authors · 2023-08-22
>
> *Experiment with more backbones*
>
> Thank you for your suggestion. We agree that the suggested experiments are valuable. However, training Transformers from scratch on large-scale video datasets and evaluation require much time considering the limited time in the rebuttal. We will try to include these results in the discussion or revised version. We note that VideoMAE/Timesformer has been widely adopted by the community, and our comparison is fair (since baselines use the same ViT Base backbone). We also compared our method with multiple other CNN models as done in SynAPT [1]. As the proposed benchmark will be released to the public, we hope the community will keep adding baselines and improving results for this important problem setting.
>
> *The reason why using the MAE compared to other self-supervised methods is not clear*
> We choose MAE because (1) it outperforms other state-of-the-art self-supervised method (eg. contrastive learning) for video action recognition [2, 3] and (2) it is data-efficient [2]. In addition, many other pre-training methods require pre-training on additional image data such as Imagenet-21K which contain additional pictures of humans. We prefer to use datasets that do not contain humans and thus adopt the MAE method.
>
> [1] Kim, Yo-whan, et al. "How Transferable are Video Representations Based on Synthetic Data?." Advances in Neural Information Processing Systems 35 (2022): 35710-35723.
>
> [2] Tong, Zhan, et al. "Videomae: Masked autoencoders are data-efficient learners for self-supervised video pre-training." Advances in neural information processing systems 35 (2022): 10078-10093.
>
> [3] Wang, Limin, et al. "Videomae v2: Scaling video masked autoencoders with dual masking." Proceedings of the IEEE/CVF Conference on Computer Vision and Pattern Recognition. 2023.

---

> > ### Comment · Reviewer_Jm2h · 2023-08-26
> > **Response to rebuttal**
> >
> > Thank you for your response. I'm satisfied with the explanations provided. While it would be beneficial to have the other backbones included in the final version, it's not mandatory.  I will keep my rating at 'Good paper, Accept'.

---

### Official Review · Reviewer_HViE · 2023-07-21
**Review on "Learning Human Action Recognition Representations Without Real Humans"**

**Rating:** 4
**Confidence:** 5
**Correctness:** Yes.
**Clarity:** Yes.

**Strengths:**

- An appropriate literature survey has been conducted, providing a good summary of the existing approaches to privacy-preserving action recognition.
- The technical approach appears sound. The developed dataset seems to be a viable solution for unsupervised or semi-supervised pre-training on a large dataset without concerns about privacy threats or data leakage.
- In contrast to previous approaches, such as altering people's appearances or blurring, the dataset contains no information about actual individuals.

**Additional Feedback:**

No additional feedback.

**Documentation:**

No.

**Ethics:**

No.

**Limitations:**

- The claim in this study that learning can be conducted without real people is largely irrelevant to privacy leakage and threat issues. In fact, training data can be sufficiently high-performing even when trained with public datasets that don't pose concerns for privacy leakage. The most significant concern regarding privacy leakage and threats arises from image/video leaks due to backdoors in computers connected to cameras during actual inference or potential network sniffing when transmitting image/video data to computing servers.
- The proposed approach seems to have no advantages over the method that simply uses synthetic data when it comes to actions in human-human interactions. Concerns about privacy-preserving may actually be more important in situations where such human-human interactions need to be distinguished.
- Overall, there is a lack of visual examples and the supplementary materials are deficient. The description of the dataset on the website also appears to be very lacking.

**Opportunities For Improvement:**

See the limitations.

**Relation To Prior Work:**

Yes.

**Summary And Contributions:**

This paper proposes a methodology for pre-training transformer models without concerns about privacy infringement or data leakage, and constructs a dataset for executing this methodology.

---

> ### Author Response · Authors · 2023-08-22
>
> *Learning without real people is largely irrelevant to privacy leakage and threat issues*
>
> We respectfully disagree with this claim. When using images of people, there are many more concerns beyond backdoor leaks and network sniffing. Most public datasets include images of people without their consent. This is not only an ethical but also a legal issue, as personal data is protected by legislation such as GDPR. Moreover, it is difficult to control for undesirable biases related to gender and race in datasets with real humans (see Hendricks et al. [1] and many other works in this area). If we consider applications of action recognition such as safety and security monitoring, removing humans from videos also prevents membership reconstruction attacks [2] that attempt to leak human identities by retrieving training data from a pretrained model. Thus, we believe it is extremely important to develop learning methods that only require virtual humans using synthetic data or human-removed real world videos.
>
> *No advantages over the method that simply uses synthetic data for human-human interactions*
>
> It is unclear which downstream task the reviewer is referring to for this claim. We want to emphasize that pretraining with no-human real data and synthetic data outperforms solely using synthetic data.  If you refer to the bottom two rows in Table 1, adding NH Kinetics to Synthetic data during MAE-Align pretraining improve the average FT accuracy by 2.5% and average LP accuracy by 5.0% across the six downstream tasks. In the paper, we explain that the improvement is because synthetic data is unable to capture the contextual bias that real world data, even with humans removed, can capture.
>
> *Lacking visual examples, supplementary materials, and  description of the dataset on the website.*
>
> Thank you for the suggestions. We included at least one visual example for each of No-Human Kinetics, Synthetic data, Kinetics, and the six downstream datasets in Figures 1, 2, and 3. We can add more examples to the final version of our paper in the appendix. The detailed version of the dataset is under the [data/](https://github.com/howardzh01/PPMA/tree/main/data) folder, which is linked in the README of [our GitHub](https://github.com/howardzh01/PPMA/tree/main). Additionally, we will add the “Datasheet for Datasets” we submitted as supplementary material to this Github.
>
> [1] Hendricks, Lisa Anne, et al. "Women also snowboard: Overcoming bias in captioning models." Proceedings of the European conference on computer vision (ECCV). 2018.
>
> [2] Hu, Hongsheng, et al. "Membership inference attacks on machine learning: A survey." ACM Computing Surveys (CSUR) 54.11s (2022): 1-37.

---

### Official Review · Reviewer_FyUv · 2023-07-22
**Learning Human Action Recognition Representations Without Real Humans Review**

**Rating:** 7
**Confidence:** 5
**Clarity:** Yes, the paper is well written and ea…

**Strengths:**

1. Privacy-Preserving Representation: The paper addresses the important issue of preserving privacy by proposing a method to pre-train action recognition models without using real human data. This is a significant contribution as privacy concerns are increasingly critical in data-driven research.
2. Novel Benchmark: The paper extends the existing SynAPT benchmark by incorporating a combination of synthetic videos and human-removed real videos. This extension allows for a more comprehensive evaluation of the proposed pre-training strategy, providing a valuable reference point for future research in the field of action recognition.
3. Closing the Performance Gap: By achieving similar performance to models trained on real-human data, the paper successfully bridges the gap between representations learned from data with and without humans. This result indicates the viability of using privacy-preserving representations in real-world applications.
4. Clear Presentation: The paper effectively communicates its objectives, methodology, and findings. The experiments are well-designed, and the results are presented in a clear and concise manner, enhancing the paper's readability and comprehension.
5. Practical Relevance: The paper's focus on privacy-preserving action recognition has real-world relevance, especially in applications involving sensitive data or scenarios where capturing real human videos might be impractical or intrusive.

**Additional Feedback:**

I really like the work proposed and the initial results. If my concerns are addressed i'm happy to further improve upon my initial review.

**Correctness:**

Yes, the dataset is constructed in a sound way and the evaluation and experiments are performed correctly.

**Documentation:**

There is sufficient detail on the collection and organization of the data. Since the primary part of the work is removal of humans, the work proposes an ethical benchmark. License has been provided explaining the terms and conditions of the use of the dataset.

**Ethics:**

No, I think the paper deals with a problem that makes action recognition datasets more ethical.

**Limitations:**

1. Missing relevant work for synthetic data training in videos: https://www.sciencedirect.com/science/article/pii/S1077314222001758?casa_token=GhH9RXYruqwAAAAA:wwllUpbOEuQlhKdYmSmRB3YUpKFl5J5s3RNkIDJqhdpd1JMT9Ym3z_NiyBeiIb8pBmvNXpNt-w , https://link.springer.com/chapter/10.1007/978-3-031-19821-2_14 , https://arxiv.org/abs/2012.03457 etc). Also studies of other forms of bias such as scene bias (https://proceedings.neurips.cc/paper_files/paper/2019/hash/ab817c9349cf9c4f6877e1894a1faa00-Abstract.html ). I think having these types of work which talk about the concepts of synthetic data training by augmentation or discussions of other forms of bias such as objects or scenes would strengthen the paper.
2. Assumptions in Synthetic Data Generation: The paper relies on synthetic data to teach the model temporal dynamics. The effectiveness of this approach may be influenced by the assumptions and quality of the synthetic data. A thorough analysis of the limitations and potential biases introduced by synthetic data generation would enhance the paper's credibility.
3. A weakness of the current work is the lack of exploration into how privacy-preserving pre-training methods scale with a larger number of videos. The paper acknowledges that this is beyond the scope of their research, but it highlights the importance of investigating scalability in future studies. Since the authors propose a data-efficient training strategy, it would be nice to actually see how this scales up with the proposed dataset with different percentages of the data being used for training.

**Opportunities For Improvement:**

1. Comparison with SOTA: The most recent relevant work for SOTA models surpass VideoMAE and Timesformer, for example mvit or mvit v2 (https://github.com/facebookresearch/mvit) , uniformer v1 and v2 etc (https://github.com/sense-x/uniformer). I think it would be good to have some of these included in the experimental analysis as these surpass the baselines suggested and would make the experimental analysis far more comprehensive.
2. Limited Discussion of Ethical Considerations: Given the focus on privacy and potential real-world applications, the paper could benefit from a more comprehensive discussion of ethical considerations related to the usage of privacy-preserving representations, especially in sensitive contexts or applications with societal impact.
3. Insufficient Insight into Failure Cases: While the proposed approach achieves promising results, the paper lacks a detailed analysis of failure cases or instances where the approach may not perform well. Understanding the limitations and failure modes of the method would provide valuable insights for future improvements.

**Relation To Prior Work:**

I have highlighted why I think the related work section needs to improved and I'm looking forward to seeing the authors response. But in terms of direct comparison to SynAPT, the difference is explained clearly.

**Summary And Contributions:**

This paper addresses the goal of pre-training representations for human action recognition without using real human data, focusing on privacy-preserving transferability. While SynAPT explored using synthetic data for this purpose, it did not effectively handle tasks with high scene-object bias. This paper proposes a solution by pre-training models using a combination of synthetic videos and human-removed real videos. The synthetic data teaches the model temporal dynamics of actions, while the human-removed real data helps it learn contextual features in the scene. This approach is evaluated on the extended SynAPT benchmark and demonstrates improved downstream task performance, closing the gap between representations trained with and without human data. The key contributions are: 1) studying the transferability of privacy-preserving representations learned from real-world videos without humans, 2) proposing an extended benchmark for evaluation, and 3) introducing a new pre-training strategy, Privacy-Preserving MAE-Align (PPMA), which significantly improves downstream task performance compared to baselines.

---

> ### Author Response · Authors · 2023-08-22
>
> Many thanks to all reviewers for their insightful review and valuable feedback! We will addess the feedback below.
>
> *Comparison with SOTA: including experiments of backbones of SOTA such as MViT and Uniformer models that  surpass VideoMAE*
>
> While we agree your suggestion would be a valuable addition, training new Transformers from scratch on large-scale video datasets was difficult due to the limited time in the rebuttal. Furthermore, to our knowledge, applying MAE to MViT and Uniformer backbone has not been tried before. Hence, it would require significant resources and time to properly conduct this experiment. We note that VideoMAE/Timesformer has been widely adopted by the community, and our comparison is fair (since baselines use the same ViT Base backbone). We also compared our method with multiple other CNN models as done in SynAPT [1] (See Table 1 of the main paper). As the proposed benchmark will be released to the public, we hope the community will keep adding baselines and improving results for this important problem setting.
>
> *Limited Discussion of Ethical Considerations*
>
> Thank you for this suggestion, we will expand on this discussion in the final version. Specifically, we will add more details about our motivation, including privacy and consent, involving not only ethical but also legal issues, as personal data is protected by legislation such as GDPR. We will add more examples related to ethics and undesirable biases related to gender and race in existing datasets (see Hendricks et al.  [2] and many other works in this area). We will also highlight other benefits of removing humans from the training data, such as preventing membership reconstruction attacks that attempt to leak human identities by retrieving training data from a pretrained model.
>
> *The paper lacks a detailed analysis of failure cases or instances where the approach may not perform well.*
>
> Our results are indeed very strong, but we will add more discussion about failure cases to the revised version of the paper. For example, for edge cases such as very crowded scenes with many people occupying the entire scene, the inpainting may be very blurry, creating noise for representation learning. Additionally, we evaluate the success of our PPMA method on the six diverse downstream tasks and compare it with the human-baseline model trained with Kinetics. As described in section 3.3, the downstream tasks range from high scene-object bias to low scene-object bias in the order of UCF101, HMDB51, Mini-SSV2, Diving48, IkeaFA, and UAV-Human. In Table 1, the PPMA model (last row) is an improvement from solely using synthetic data (2nd to last row) but does worse than the human-baseline model (first row) on three of the six tasks – specifically, high scene-object tasks such as UCF101, HMDB51, and Mini-SSV2. While training on No-Human Kinetics as an addition to Synthetic Data helps the model learn scene-object relations, there is still a small gap with the human-baseline for the high scene-object tasks.
>
> *Missing relevant work for synthetic data training in videos*
>
> Thank you for suggesting additional relevant work about the overreliance on the background features, which we will add to the related works section.The common theme among VideoMix [3], Learn2Augment [4], and ActorCutMix [5] data augmentation methods are that they swap the action and background of two videos. Additionally, Choi et. al 2019 [6] proposes to add an adversarial loss to ensure the model does not over rely on background features. While we also separate the human and the background for the No-Human Kinetics dataset, our method is different because we learn background features from real data and temporal information from synthetic data. Thus, when we generalize to new downstream tasks, the model has priors about background features but can also focus on temporal action features.
>
> *Additional analysis on Assumptions in Synthetic Data Generation*
>
> While the reviewer mentions an excellent point in that synthetic data can allow for various studies through control over different bias variables, we emphasize that we experimented with a pre-generated fixed-size synthetic dataset from SynAPT. While the bias information of this data is available in SynAPT [1] and its sources, one of its primary aspects is that it de-correlated actions and backgrounds, making the model trained on it focus on temporal features. This is a property often lacking in real videos where the actions may be correlated to the contexts they take place in.
>
> *Lack of exploration into how privacy-preserving pre-training methods scale with a larger number of videos*
>
> Thank you for your suggestion. For fair comparison, we followed the specifications from SynAPT [1] including dataset size. We agree that these will be valuable experiments and would be a useful expansion for future work.

---

> > ### Author Response · Authors · 2023-08-22
> > **Citations**
> >
> > Citations included below due to character limit.
> >
> > [1] Kim, Yo-whan, et al. "How Transferable are Video Representations Based on Synthetic Data?." Advances in Neural Information Processing Systems 35 (2022): 35710-35723.
> >
> > [2] Hendricks, Lisa Anne, et al. "Women also snowboard: Overcoming bias in captioning models." Proceedings of the European conference on computer vision (ECCV). 2018.
> >
> >  [3] Yun, Sangdoo, et al. "Videomix: Rethinking data augmentation for video classification." arXiv preprint arXiv:2012.03457 (2020).
> >
> > [4] Gowda, Shreyank N., et al. "Learn2augment: learning to composite videos for data augmentation in action recognition." European conference on computer vision. Cham: Springer Nature Switzerland, 2022.
> >
> > [5] Zou, Yuliang, et al. "Learning representational invariances for data-efficient action recognition." Computer Vision and Image Understanding 227 (2023): 103597.
> >
> > [6] Choi, Jinwoo, et al. "Why can't i dance in the mall? learning to mitigate scene bias in action recognition." Advances in Neural Information Processing Systems 32 (2019).

---

> > > ### Comment · Reviewer_FyUv · 2023-08-24
> > > **Response to rebuttal**
> > >
> > > I'm happy with the response from the authors and my concerns would mostly be solved if the content promised is added. Neurips provides the authors the option to edit the paper and reupload a version that shows how the reviewer's concerns have been addressed and I think the author's should do that so we can see how our concerns are being addressed.

---

> > > > ### Author Response · Authors · 2023-08-26
> > > >
> > > > We've updated the paper. Please see the general response to reviewers for details of revisions made. Thank you!

---

### Author Response · Authors · 2023-08-26
**Uploaded Revision based on Reviewer Feedback**

Thank you all reviewers for the insightful feedback. We revised the paper and appendix based on the feedback, with edits being in blue font, and made it available for download on the OpenReview site.

See below for a list of the edits we made and where in the paper we revised based on the reviewers feedback.

Reviewer 1:
- *Limited Discussion of Ethical Considerations*  (see Introduction line 39-45)

- *The paper lacks a detailed analysis of failure cases or instances where the approach may not perform well.* (included in 4.2)

- *Additional analysis on Assumptions in Synthetic Data Generation* (included in Sec 3.2)

- *Missing relevant work for synthetic data training in videos* (included in Sec 2 Related Works)

Reviewer 2:

- *Lacking visual examples, supplementary materials, and  description of the dataset on the website.* (see Appendix B)

Reviewer 3:

- *The reason why using the MAE compared to other self-supervised methods is not clear* (see Sec 3.4)

Reviewer 4:

- *More analyses should be added to support the claim entirely. For example, how to ensure the segmentation and inpainting quality in the first step? What is the influence of artificial inpaintings?* (see Appendix b)

- *The data contribution is a little marginal, with only the real data processing and without manual checking.* (see Appendix B)

- *More details of the models and pre-training should be given to determine the factors affecting the performance.* (see Appendix A)

- *Tab. 4: please add the model names in the ratio places.* (done. See table 4)

---

### Decision · Program_Chairs · 2023-09-22

**Decision:**

Accept (Poster)

**Comment:**

Most of the reviewers are positive about the paper and recognize its merits. The ACs read through the negative review and believe the authors address the concerns properly.